# Optimized Method for the Identification of Candidate Genes and Molecular Maker Development Related to Drought Tolerance in Oil Palm (*Elaeis guineensis* Jacq.)

**DOI:** 10.3390/plants11172317

**Published:** 2022-09-04

**Authors:** Sunya Nuanlaong, Suwit Wuthisuthimethavee, Azzreena Mohamad Azzeme, Potjamarn Suraninpong

**Affiliations:** 1School of Agricultural Technology and Food Industry, Walailak University, Nakhon Si Thammarat 80161, Thailand; 2Faculty of Biotechnology and Biomolecular Sciences, Universiti Putra Malaysia, Serdang 43400, Malaysia; 3Biomass and Oil Palm Center of Excellence, Walailak University, Nakhon Si Thammarat 80161, Thailand

**Keywords:** water deficit, next-generation sequencing, EST-SSR

## Abstract

Drought is a major constraint in oil palm (*Elaeis guineensis* Jacq.) production. As oil palm breeding takes a long time, molecular markers of genes related to drought tolerance characteristics were developed for effective selection. Two methods of gene identification associated with drought, differential display reverse transcription polymerase chain reaction (DDRT-PCR) and pyrosequencing platform, were conducted before developing the EST-SSR marker. By DDRT-PCR, fourteen out of twenty-four primer combinations yielded the polymorphism in leaf as 77.66% and root as 96.09%, respectively. BLASTN and BLASTX revealed nucleotides from 8 out of 236 different banding similarities to genes associated with drought stress. Five out of eight genes gave a similarity with our pyrosequencing sequencing database. Furthermore, pyrosequencing analysis of two oil palm libraries, drought-tolerant, and drought sensitive, found 117 proteins associated with drought tolerance. Thirteen out of sixty EST-SSR primers could be distinguished in 119 oil palm parents in our breeding program. All of our found genes revealed an ability to develop as a molecular marker for drought tolerance. However, the function of the validated genes on drought response in oil palm must be evaluated.

## 1. Introduction

Oil palm (*Elaeis guineensis*) is a perennial monocotyledonous crop plant producing the most oil in the world; its production is equal to 10 t ha-1 or 13 t ha-1 year-1 palm oil under favorable conditions [1]. Due to climate changes and limitations of agricultural irrigation water caused by global warming, oil palm production is limited in several areas in southern Thailand. Drought is one of the most severe problems for oil palm [2]. The effects of a drought period on yield components of oil palm occur with long time lags because drought-sensitive processes such as floral sex determination or early inflorescence abortion occur months or years before the maturity of a given bunch [3]. Therefore, drought tolerance is one of the most critical goals of oil palm breeding programs. Nonetheless, drought tolerance is complicated and still not fully understood.

Improving drought tolerance in oil palm varieties with classical plant breeding approaches is slow, time-consuming, and costly [4]. During the last two decades, many scientists have reported that breeding for tolerance to stress factors was possible using molecular markers [5]. The discovery of new genes involved in water stress response and/or tolerance before molecular marker development appears to be a crucial step. Currently, transcriptome analysis with next-generation DNA sequencing (NGS) technology has been utilized under various conditions, such as environment, cell type, development stage, and cell state [6]. It can assist in identifying target genes and the variation of their expression by exploring mRNA expression difference and function annotation [7].

Moreover, the transcriptome is a precious resource for the discovery and identification of a polymorphic molecular marker, such as sequence repeats (SSRs) and single nucleotide polymorphisms (SNPs) [8]. Pyrosequencing via Roche’s 454 GSFLX platform is one of the valuable tools for discovering novel transcripts in many plant species [9,10]. It can discover genes of unknown function, sequences with high-quality base discrepancies, and alternative splice variants [11]. An mRNA differential display (DDRT-PCR), a gel-based transcript profiling system designed for analyzing differences in gene expression levels, is simple, quick, sensitive, and powerful for screening cDNA [12,13]. It is widely used in plant systems to study specific gene expression, such as genes involved in stress response [14], fruit ripening [15], and plant-pathogen interaction [16].

Many efforts have elucidated the mechanisms of drought tolerance in plants through genomics approaches, and several genes associated with drought stress have been reported [17,18,19,20]. In oil palm, transcriptome profiling of root [20] and the transcription factor of leaf, and WRKY, under drought stress, have been reported [21]. In addition, genome-wide SNP marker identification of fresh fruit bunch (FFB) production associated with drought tolerance in high and low-yielding oil palm was validated. However, transcriptional changes responding to extended periods of drought stress have yet to be identified. Moreover, a limitation of comparing differences in transcription levels between drought-tolerant and drought-sensitive oil palm genotypes under drought stress conditions was not previously reported. Therefore, this study conducted sequencing analysis via pyrosequencing and DDRT-PCR of drought-tolerant and drought-sensitive oil palm genotypes under drought stress. Furthermore, our breeding program developed and screened an SSR marker with an oil palm parent.

## 2. Results

### 2.1. Differential Display Reverse Transcriptase PCR (DDRT-PCR) and RT-PCR

The total RNA from four oil palm varieties had high purity suitable for cDNA synthesis. The PCR reaction with 18 s rRNA confirmed cDNA quality as a size of 530 bp. Twenty-three out of twenty-four primer combinations could be amplified, yielding a product size greater than 10 bp. However, after separating on 6% polyacrylamide gel, fourteen out of twenty-three primer combinations provided the different banding as 1326 bands and 1359 bands for leaf and root, respectively. The DNA banding was between 200 and 1200 bp, with the polymorphism in the leaf at 77.66% and the root at 96.09% (Table 1). After a random cut of 236 bands, the different banding on the gel was observed; 81 bands could be amplified with the same primer combinations. The result from BLASTN revealed 60% similarity of 72–99% of 25 DNA bandings associated with the nucleotide sequence in the NCBI database. These sequencings were similar to *Phoenix dactylifera*, *Elaeis guineensis*, and others at 64%, 16%, and 20%, respectively. Eight out of twenty-five sequences had similarities to genes associated with stress conditions (Appendix A).

Moreover, the nucleotide sequences of all 81 bands, when aligned with BLASTX to compare protein translation as the similarity with more than 35% confidence, found similarities ranging from 42 to 99%, with 19 proteins in the NCBI database. Eight of the nineteen proteins were associated with stress responses in plants. However, the other 11 proteins are unknown and unrelated to a stress response. They are involved in cell division, adenylate cyclase activity, DNA integration, transferase activity, and peroxisome (Appendix A).

### 2.2. 454 Sequencings, Transcriptome Assembly, and Annotation

The total sequence output of Lamé and Surathani 1 were 165,872 reads, totaling 61.4 Mb and 144,592 reads, respectively. The assembled data of Lamé produced 4100 contigs, 5358 isotigs, and 4493 isogroups. However, Surathani 1 produced 3171 contigs, 4193 isotigs, and 3613 isogroups. The N50 of Lamé contig size in the combined assembly was 856 bp, larger than Surathani 1, having a contig size of 839 bp. The obtained sequences were BLASTX with NCBI database examining genes associated with drought stress. Lamé and Surathani 1′s sequences as 15.70% (20,040 reads) and 18.82% (22,870 reads) could be translated to protein, respectively. The mapped proteins were both associated and non-associated with the dehydrated condition. After filtrate, 117 proteins were associated with drought tolerance, which could be separated into 11 biochemical groups as follows: (1) abscisic acid, (2) antioxidant, (3) compatible compound, (4) ethylene, (5) heat shock protein, (6) helicase, (7) mannitol, (8) proline, (9) senescence/ripening, (10) transcription factor, and (11) trehalose (Appendix A).

Repeated sequence forms of two transcriptome libraries were examined to develop EST-SSR markers associated with drought tolerance. Furthermore, 227 repeated sequence positions from 226 genes and 201 repeated sequence positions from 191 genes were generated from the drought-tolerant and drought-sensitive library after dehydrating for 45 days, respectively. When comparing redundancy genes between both libraries and displayed in the Venn diagram, 165 genes were found in the drought-tolerant library, 121 in the drought-sensitive library, and 48 in both libraries. However, after rechecking, 23 genes were replicated in the drought-tolerant library. Finally, 142 genes were used for Gene Ontology. Further, 136 out of 142 genes were mapped with the CloudBlast database, 92 with the GO database, and 79 genes could be annotated. In addition, most E-values ranged from 1 × 10^−5^ to 1 × 10^−50^ (37%), followed by 1 × 10^−51^ to 1 × 10^−100^ (35%), and 1 × 10^−151^ to 1 × 10^−180^ (11%) (Figure 1a). The nucleotide sequences had 92% similarity to the nucleotide sequences of oil palm (*Elaeis guineensis*), followed by date palm (*Phoenix dactylifera*) at 4%, and the other at 4%, respectively (Figure 1b). Gene Ontology analysis of repeated genes at GO level 2 could classify functional groups into three: biological processes, molecular function, and cellular components (Appendix A). 

Further, functional analysis of proteins obtained from the InterPro Scan technique and InterPro protein signature databases found 142 genes out of 192 protein functions. The twenty top-hit were the proteins in domain function. Thioredoxin-like fold (domain) (IPR012336) was the highest gene (five genes), followed by thioredoxin domain (domain) (IPR013766) (three genes), AP2/ERF domain (domain) (IPR016177) (three genes), and DNA-binding domain (domain) (IPR015943) (three genes), respectively (Figure 2). The analysis of protein function with the PANTHER database (http://pantherdb.org/, accessed on 28 May 2020) also found the largest functional group of thioredoxin (PTHR10438), similar to the analysis with InterPro protein signature databases.

Moreover, KEGG analysis involved 11 major metabolic pathways, 31 secondary pathways, and 15 enzymes. Lipid metabolism was the highest number of enzymes and the highest number of genes (10 enzymes, 6 genes), followed by amino acid metabolism (8 enzymes, 5 genes) and carbohydrate metabolism (7 enzymes, 5 enzymes) (Figure 3).

### 2.3. Comparison of the Expressed Gene Associated with the Drought Tolerance between DDRT-RT and Pyrosequencing and Real-Time PCR

All eight genes were mapped with both MPOB and NCBI databases. However, five of eight genes from DDRT-PCR were mapped with a transcriptome sequencing database. They contained histone H2A (99% similarity), cysteine proteinase (Cys) (100% similarity), pentatricopeptide repeat-containing protein (32% similarity), trehalose-6-phosphate synthase (87.5% similarity), and serine/threonine-protein phosphatase PP1 (92% similarity) (Appendix A). Nevertheless, the other three genes were only found in DDRT-PCR sequencing. Then, two genes, ATP-dependent DNA helicase Pif-1-like, and bHLH106 expressed only in DDRT-PCR, and two genes, histone H2A and cysteine proteinase expressed in both sequencing methods were selected for RT-PCR. PIF1 helicase, a putative gene, demonstrated the highest expression in a leaf of oil palm variety Surathani 2 after a water deficit for 30 days (1.7 times). However, in the oil palm variety Surathani 1, low, relatively consistent gene expression through water deficit times was found (Figure 4). The expression of transcription factor bHLH106-like gene depicted the highest expression in a leaf of oil palm variety Ghana after water deficit for 30 days (0.53 times), non-significantly differ from oil palm variety Lamé having water deficit for 60 and 75 days. Nonetheless, the oil palm variety Surathani 1 revealed the lowest expression value in leaves through water deficit times (Figure 5). The expression of histone H2A illustrated the highest expression in a leaf of oil palm variety Surathani 2 after water deficit for 45 days (2.05 times) with non-significant difference from oil palm variety Lamé after water deficit for 45 and 60 days (1.86 and 1.83 times, respectively) (Figure 6). The expression of Cys revealed the highest expression in the root of oil palm variety Surathani 1 after a water deficit for 15 days (8.82 times). However, low expression of this gene was found in oil palm variety Lamé after a water deficit exceeding 15 days (Figure 7).

### 2.4. Molecular Marker Development

One hundred and forty-two genes were used for sequence distribution analysis for molecular maker development via the EST-SSR technique. The highest and the lowest repetitions were 3-base and 5-base, respectively (Appendix A). Then, 142 genes with 125 positions were designed for 60 primers. Furthermore, 44 out of 60 primers could be amplified with an annealing temperature ranging from 56 to 65 °C. After amplifying the obtained primer with every five samples of drought-resistant and drought-sensitive, 23 out of 44 primers revealed polymorphic bands. However, the left of the primer depicted a monomorphic band. However, the analysis of allele and genotype frequencies revealed 5 out of 23 primers, primer isotig0310, isotig03937, isotig04263, isotig04783, and isotig05050, distinctly distinguished the differentiation between drought-tolerant and drought-sensitive varieties (Appendix A). All alleles were variated in the range of 2–5 alleles per primer with 11 genotypes. Primer isotig03937 is the most clearly identified drought stress variety. The drought-tolerant variety showed an A allele with the AA genotype. In contrast, the drought-tolerant variety depicted the B allele with the genotype of BB. Nevertheless, the other primer revealed alleles A, B, and C (Appendix A). Moreover, 10 out of 13 primers could be aligned with the oil palm reference genome (Appendix A).

Moreover, 23 primers were tested with 119 oil palm parents in our breeding program. All primers were able to distinguish between drought-tolerant and drought-sensitive oil palms. The genetic correlation of oil palm parents had a coefficient of 0.57, divided oil palm samples into five groups with three independent samples (S48, S53, S121) (Figure 8A). The oil palms belonging to Group 2 and Subgroup 2 were genetically closely related to the L sample (S120, Lamé variety), especially oil palm numbers S17, S97, S40, S42, S52, S112, S105, S108, S27, S28, S41, S43, S44, S51, and S47 (Figure 8B).

## 3. Discussion

The identification of the drought-tolerant gene in oil palm via DDRT-PCR and pyrosequencing technology was conducted in this study. DDRT-PCR revealed polymorphisms at cDNA levels in assuming drought at different time periods when compared between the leaf and root of four oil palm varieties with twenty-four primer combinations. Meanwhile, the different periods of drought stress generated genetic changes among the tissue and oil palm varieties. However, pyrosequencing technology provided more details of the genes involved in the drought trial between the four oil palm varieties, tolerance and sensitivity, and the analysis of the different gene expressions between both trials, resulting in interesting genes that could be used for molecular marker development. By using these two techniques, 8 genes from DDRT-PCR and 117 proteins from pyrosequencing were associated with drought tolerance. By these, five of eight genes from DDRT-PCR were mapped with the gene in the pyrosequencing database (Appendix A). These eight genes were the key dominant drought-tolerant genes in our study. The expression of these eight genes was similar to genes that were previously reported in drought conditions in many plant species, except for ATP-dependent DNA helicase Pif1--like. The expression of histone H2A was similar to *Arabidopsis*, and it was found that the overexpression of the H2A variant gene TaH2A.7 from wheat significantly enhanced drought tolerance [22]. The suppression of cysteine proteases in wheat drought-tolerant cultivar was reported by [23]. The expression of PPR was similar to the drought tolerance response in many plant species [24,25,26], as this gene plays a key role in various biological processes in plants due to its involvement in ABA signaling [27]. The greater expression of trehalose-6-phosphate genes allows plants to increase drought-tolerant conditions by controlling the changes of the ABA signaling cascade to trigger stomatal closure and glucose signaling during seedling growth under dehydration stress conditions [28,29,30]. Serine/threonine-protein phosphatase PPP1 plays an important role in the response to stress conditions by regulating the signal transduction of phytohormone in the immune system such as ethylene, salicylic acid, jasmonic acid, and other derivatives [31]. The basic helix–loop–helix (bHLH) plays an important role in signal reception and transmission. After plants are affected by drought stress, this gene induces drought responses by regulating stomatal development and stomatal density, regulating hormone metabolism and ABA signaling processes involved in the formation of trichrome and root hair [32]. The formation of trichrome, together with the number of stomata, was observed in our previous study [33]. ATP-dependent DNA helicase Pif1--like was reported to control plant growth under stress conditions by controlling the stress response mechanism [34,35,36], especially wounding stress [37]. There is no previous report on drought tolerance. It is probable that LRR receptor-like serine/threonine-protein kinase At3g 47570-like is involved in drought resistance by acting as a positive regulator of ABA signaling [38,39]. From our results obtained, Lamé and Surathani 2 was confirmed as a drought-tolerant variety, while Ghana and Surathani 1 was confirmed as a drought-sensitive variety. However, our findings of drought-tolerant genes differed from those of previous findings in peanuts [40], chickpeas [41], corn [42], rice [43], and *Ruta graveolens* [15]. Further, our founding genes were used for the molecular marker development of drought stress.

For 454 pyrosequencing, two sstDNA libraries were shorter than the results obtained from pepper (*Capsicum annuum* L.). Still, the total sequence data were longer than pepper [44] with the same technology. Because we conducted ½ 454 pyrosequencing, some transcripts may have been lost. The low-expression abundance affected the expression patterns based on raw reads data reflecting lower members of reads in our library. Additionally, BALSTX revealed 15.70% (Lamé) and 18.82% (Surathani 1) of the trimming sequences, respectively. As less than 32,000 assembled sequences were matched with unknown or known proteins, the short sequencing reads obtained using next-generation technology, the non-hits of a likely fraction of sequences, including alternative splice variants, novel gene products, and differentially expressed genes. were reported to detect significant sequence similarity [45]. Moreover, 117 filtrated proteins were organized into 11 biochemical groups associated with drought stress conditions in plant species [46,47,48,49,50,51,52]. Interestingly, the WRKY transcription factor was found in drought-tolerant and drought-sensitive libraries with different types and copy numbers. A study was reported in Group 3 of *E. guineensis* WRYK transcription factors, including EgWRKY07, 26, 40, 52, 59, 73, and 81, which displayed multiple roles in the drought response of the oil palm seedling leaf [22]. However, in African oil palm (*E. guineensis*, pisifera), EgWRKY07 and EgWRKY52, with high similarity to AtWRKY30 in Arabidopsis, it was reported that they are strongly upregulated to salinity stress [53]. Conversely, our study found the expression of different WRKY transcription factors, such as WRKY transcription factor putative, WRKY transcription factor 7, WRKY transcription factor 20, WRKY transcription factor 24, WRKY transcription factor 27, and WRKY domain transcription factor. Thus, the expression of the WRKY transcription factor from the previous study and our study might indicate the crucial roles of these genes in oil palm in response to drought stress and other abiotic stresses. However, further examination of WRKY transcription factors in our study must be conducted to determine their response to abiotic stresses and molecular marker development.

Further, our study’s highest process of each GO term was similar to *Ammopiptanthus mongolicus* root and leaves under drought stress [11,54], respectively. This was also the case in *Prunus sibirica* L. leaves under drought stress [55] when analyzed via the 454 pyrosequencing method. However, it differs from the root of four *Gossypium herbaceum* genotypes when analyzed with the same technique [56]. Contrarily, the oil palm seedling root transcriptome on Day 14 under drought stress revealed the most GO terms related to cell wall biogenesis and functions. This was followed by phenylpropanoid biosynthesis, metabolism, ion transport, homeostasis, and response to osmotic stress and water homeostasis [57]. The difference in gene expression of drought tolerance from previous studies might result from sequencing technology, plant stage, and plant variety. Nevertheless, our findings on thioredoxin revealed similarity with the upregulation of thioredoxins (TRXs) in rice [58] and *Arabidopsis* under drought conditions [59]. Further, the highest number of enzymes and the highest number of genes in the lipid metabolism, followed by amino acid metabolism and carbohydrate metabolism in our KEGG pathway, slightly differed from the KEGG pathway analysis of *A. mongolicus* drought tolerance. It was found that metabolic pathways had the highest number of enzymes, followed by ribosomes and the biosynthesis of secondary metabolites [11]. In contrast, in rice seedlings, drought stress revealed plant hormone signal transduction as the most enriched metabolic pathway, followed by carbohydrate metabolism-related pathways [60]. However, carbohydrates, amino acids, lipids, and energy metabolism are involved in the drought response of poplar trees [61] and soybean [57].

For EST-SSR, 142 genes were screened. Because 1 gene consisted of 2 to 3 isotigs, 125 positions obtained from 142 genes were used for primer design. Among the 60 primers designed, most tri-nucleotides revealed the most frequent type of SSR motif. This finding aligns with the results reported in cereals [62], *A. mongolicus* [11], and *C. annuum* [44]. Five out of twenty-three primers were clearly distinguished between the drought-tolerant and drought-sensitive varieties. Notably, primer isotig03937 was the most clearly identified drought stress variety. Primer isotig03937 was mapped with spliceosome-associated protein 130 A (*E. guineensis*). Interestingly, this gene had no previous report on plant species’ drought tolerance or abiotic response. A further study must be conducted on this gene’s abiotic response. Additionally, 13 primers tested with 10-year-old oil palm parents in our breeding program revealed a close genetic relation to Lamé. Thirty percent of oil palm samples in this group demonstrated bunch production collaboration with drought tolerance markers. Moreover, drought stress is controlled by genetics, the environment, and interaction between the two with plants of different ages. Thus, some markers obtained from our study could not accurately determine drought stress for qualitative characteristics. Further, our new finding genes and more molecular markers must be determined.

## 4. Materials and Methods

### 4.1. Plant Materials and Dehydrated Condition

Four oil palm commercial varieties at 12 months old, Lamé, Surathani 2 (Deli × Lamé), Ghana, and Surathani 1 (Deli × Calabar), were grown in composite soil (sand, soil, and coconut husk at 1:1:1) in 200L fiberglass tank in the greenhouse at 28 ± 2 °C with the relative humidity nearly 80% at Walailak University, Nakhon Si Thammarat, Thailand. The oil palm variety, Lamé, is known for its drought tolerance. In contrast, the Ghana variety is known for its drought sensitivity, both obtained from CPT-Agrotech Co., LTD, Chumporn province, Thailand. The oil palm variety Surathani 1 is expected to be well-watering. At the same time, the variety Surathani 2 is known for its drought tolerance. Both were obtained from Surathani Oil Palm Research Center, Surathani province, Thailand. Well-watering oil palm plants conducted water stress until the tensiometer reached 0 centibars, then irrigation stopped until 90 days. This experiment used five oil palm plants for irrigation and no irrigation.

On the contrary, for control, well-watering was performed to keep the tensiometer at 45 centibars for 90 days. For DDRT-PCR, leaf and root tissue were harvested at 15-day intervals until 90 days. Leaf samples were collected from the second leaf stalk. Conversely, root samples were collected from the adventitious root at 15–20 cm depth from the soil. For the pyrosequencing study, the second leaf of Lamé and Surathani 1 at 45 days after dehydration was collected for RNA extraction. All samples were immediately placed in liquid nitrogen and kept at −80 °C until usage.

### 4.2. Differential Display Reverse Transcription Polymerase Chain Reaction (DDRT-PCR)

The total RNA at different dehydration times was extracted according to the instructions of the Total RNA Mini Kit Plant (Geneaid, Taiwan). The number of replications was three. The successful RNA extraction was confirmed with Eppendorf BioPhotometer Plus Model #6132 (Eppendorf, Germany) and 1% agarose gel electrophoresis. Then, RNA was transcribed to cDNA using a Transcriptor High Fidelity cDNA Synthesis Kit (Roche, Germany), followed by the instructions. A total of 20 μL cDNA synthetic reaction consisted of 2 μL total RNA (2 μg), 1.0 μL oligo (dT) primer (2.5 μmolar), 4.0 μL reaction buffer, 0.5 μL protector RNase inhibitor (40 unit/μL), 2.0 μL deoxynucleotide mix (dNTP), 1.0 μL reverse transcriptase, and 8.5 μL sterile dH2O. The reaction was performed in Eppendorf (T100 Thermal Cycler, Bio-Red, USA) by incubating at 65 °C for 10 min, followed by quick chilling on ice until it cools off, then continuous incubating at 45 °C for 30 min, 85 °C for 5 min, and terminated by heating at 95 °C for 5 min. All cDNA samples were stored at −80 °C until further use. For qualitative examination, the obtained cDNA was amplified with 18 S rRNA under the following conditions: 95 °C/4 min, 95 °C/30 s, 55 °C/30 s, 75 °C/1 min for 35 cycles, 75 °C/7 min, and 4 °C before running on 1.5% agarose gel.

Differential Display RT-PCR (DDRT) was performed as described by [63] with minor modifications. A total of four anchors and six arbitrary primers (Appendix A) were used. The amplification was conducted in a final reaction volume of 10 μL containing 1 μL each cDNA as template, 5 μL Toptaq Master Mix Kit buffer (QIAGEN, Germany), 0.5 μL 2.5 μmol anchored oligo (dT) primer, 0.5 μL 2.5 μmol arbitrary primer, and 3 μL sterile dH2O. The amplification program included the protocol of 94 °C/3 min, 40 cycles of 94 °C/30 s, 50 °C to 58 °C (according to primer combination)/60 s, and 72 °C/15 min for a final extension. Each reaction was run in triplicate. The products were denatured with 6% polyacrylamide gels and dried with silver nitrate before visualizing banding. Further, the selected primer combinations were amplified for all samples. After analyzing their genetic diversity, DDRT banding was excised randomization, eluted, reamplified, and purified via QIAquick PCR Purified Kit (QIAGEN, USA). After rechecking the banding size on 1.5% agarose gel, the DNA was ligated into the pGEM-T Easy vector (Promega, USA) before transforming into DH108 via Multiporator (Eppendorf, Germany). The DNA of selected plasmids was extracted and purified with the High Pure Plasmid DNA Isolation Kit (Merck, Germany) manually followed before being sequenced at Pacific Science Co., LTD, Thailand. The nucleotide sequence of all clones was compared with the sequences deposited in GenBank using BLASTN and BLASTX (http://www.ncbi.nlm.nih.gov/BLAST/, accessed on 16 June 2019). Sequence alignments were performed with CLUSTAL_X.

### 4.3. RNA Isolation, Transcriptome Analysis, and Sequence Annotation

The total RNA was extracted from the leaf tissue of each sample at 45 days to monitor the molecular response of water deficiency. A 3–5 g leaf was ground in liquid nitrogen before putting the fine powder into a 50 mL tube, 25 mL extraction buffer (5 M guanidium isothiocyanate, 31 mM sodium acetate pH 8.0, 1% β-mercaptoethanol, 0.88% sacosin, and 1% polyvinylpyrrolidone) was added and heavily shaken before being kept in ice for 10 min. The mixture was centrifuged at 4000× *g*, 4 °C for 45 min; the supernatant was put into an ultracentrifuge tube, 5.7 M cesium chloride was added and centrifuged in an ultracentrifuge (himac CP100WX, Hitachi Koki Co., Ltd., Tokyo, Japan) at 25,000 rpm, 20 °C for 20 hr. After discarding the supernatant, the pellet was washed with 70% ethanol, dried at room temperature for 1 h, dissolved with 200 ul RNase-free water, put on ice for 1 h, and homogenized regularly. The total RNA was transferred into a new 1.5 mL tube and kept at −80 °C before usage. The mRNA was removed from the extracted RNA via PolyATtract^®^ mRNA Isolation System I and II (Promaga, Madison, WI, USA) following the manual’s instructions.

The GS- FLX Titanium, the 454 Life Sciences sequencing platform (Roche, Indianapolis, IN, USA), discovered the nucleotide sequence of an interested mRNA. RNA-sequence data from each library were assembled with 454/Roche’s newbler v2.6. The water stress variety (Lamé) assembly library was compared with the water stress sensitive (Surathani 1) library. Then, the combined group of both samples was annotated using Blast2GO against the plant non-redundant database. Gene Ontology distribution was assigned using the Blast2GO function [64] and reduced isogroups with more than one isotig to a single isotig representative of the group for GO analysis to avoid overrepresentation of isogroups with multiple isotigs. BioVenn (http://www.cmbi.ru.nl/cdd/biovenn/, accessed on 6 June 2020) and area-proportional Venn diagrams [65] were used to summarize the overlap of the gene expressed between two oil palm varieties under water deficit conditions. Blast2GO function was used to analyze the KEGG pathway in the transcriptome of two oil palm varieties under water deficit conditions. For biological interpretation of higher-level systemic functions, the KEGG pathway mapping is mapping molecular datasets, especially large-scale datasets in transcriptomics [66].

### 4.4. Comparison of the Expressed Gene Associated with Drought Tolerance from Transcriptome and DDRT-RT and Real-Time PCR

The nucleotide sequence from BLASTx of DDRT was aligned with the isotigs from the pyrosequencing and searched against NCBI and MPOB databases before being selected for the RT-PCR study. Primers of those selected genes were designed using Primer3 Input (http://bioinfo.ut.ee/primer30.4.0/, accessed on 8 November 2020) and Oligo-Nucleotide Properties Calculator (http://www.basic.northwestern.edu/-biotools/-oligcalc.html, accessed on 8 November 2020). The primer was designed as 18–23 bp with the PCR product size of 200–300 bp, 40–60% CG content, and 50–70 °C salt adjusted with both forward (F) and reverse (R). The amplified reaction of 10 μL contained 1.0 μL cDNA (diluted 10 times), 5 μL Toptaq buffer, 0.25 μL oligo (dT) primer (1.0 μmolar), 0.25 μL arbitrary primer (1.0 μmolar), and 3.5 μL sterile dH2O. Gene expression was performed via Real-Time PCR (7300 Real-Time PCR System (Bio-Rad, Hercules, CA, USA) by using EvaGreeen® Super Mix (Bio-Rad, Hercules, CA, USA) as a supermix. 18 S rRNA was used as a reference gene. The RNA from the leaf and root at 0, 15, 30, 45, 60, 75, and 90 days after dehydration was used. The PCR condition was as follows; 1 cycle at 95 °C/3 min, 95°C/30 s, 60 °C/30 s for 35 cycles, 72 °C/7 min, and keep at 4 °C. The triplet replication was used with 18 s rRNA as a reference with the standard DNA as 10^−1^, 10^−2^, 10^−3^, and 10^−4^ times. A melting curve was used to confirm the PCR product. Expression data were analyzed as cited [67].

### 4.5. Single Sequence Repeat (SSR) Molecular Marker Development

A repeating base sequence was searched from the pyrosequencing database for the SSR marker. The expressed sequence tag (EST) under dehydrated conditions of the drought-tolerant and drought-sensitive library was set for repetition base as two bases (10 replications), three bases (6 replications), four bases (5 repetitions), five bases (4 repetitions), and six bases (3 repetitions) via WebSat (http://wsmartins.net/websat/, accessed on 10 November 2020). The repeated sequences were compared for primer design, and genes were selected only in drought-tolerant libraries. Sixty primer pairs were designed from 40 sequences of replicated genes using Primer3 Plus and the Oligonucleotide Properties Calculator program. After scanning PCR conditions with Surathani 2, the primers were amplified with 119 oil palm parents under a CPI Agrotech Co., Ltd., Thailand breeding project. The volume of all primer reactions was set as 10 μL; 2 μL DNA, 5 μL Toptaq buffer, 0.25 μL primer forward, 0.25 μL primer reverse, and 2.5 μL sterile dH2O. Gradient temperatures annealing were performed for suitable PCR condition: 94 °C/3 s, 94 °C/30 s, 40 cycles at 40–60 °C/30 s, 72 °C/60 s, and 72 °C/7 min before being kept at 4 °C. The PCR products were run in 1% agarose gel for 30 min and dyed with ethidium bromide before being counted and calculated to distinguish the genetic diversity of all oil palm samples.

## 5. Conclusions

Drought is one of the natural phenomena affecting yield in oil palm, considered a quantitative trait involving the participation of a complex set of genes. In this study, we produced some new transcriptomic information for oil palm drought stress via DDRT-PCR and 454 sequencing technology for further marker development in our oil palm breeding program. DDRT-PCR exposed eight genes associated with drought stress in plants. In contrast, the pyrosequencing of two libraries, drought-tolerant and drought-sensitive varieties, revealed 117 proteins related to drought tolerance. The drought tolerance library had a repeated sequence of 142 genes for annotation with GO ontology, InterPro Scan technique, InterPro protein signature database analysis, and KEGG analysis. Five of eight genes, namely histone H2A, cysteine proteinase, pentatricopeptide repeat-containing protein, trehalose-6-phosphate synthase, and serine/threonine-protein phosphatase PP1, from DDRT-PCR were mapped with transcriptome sequencing database. In contrast, bHLH106, ATP-dependent DNA helicase PIF1-like, and probable LRR receptor-like serine/threonine-protein kinase At3g47570 were found in DDRT-PCR only. Notably, the WRKY transcription factor in our pyrosequencing data differed from that reported in oil palm seedlings under drought stress. These genes, together with spliceosome-associated protein 130 A (*E. guineensis*), similar to primers isotig03937, are engaging for further study in response to drought and abiotic stress in oil palm. Moreover, 5 of 60 EST-SSR primers were identified as drought-tolerant and drought-sensitive varieties. Although 23 primers tested with 119 oil palm parents could separate the oil palm sample into 5 groups with Group 2 and Subgroup 2, genetically closely related to the drought tolerance sample, only 30% of brunch production in these groups responded to our marker. In parallel, identifying target gene expression using DDRT-PCR and 454 sequencing technology is considered a potential procedure for gene selection in improving the tolerance of oil palm trees. Our result might be the basis for an in-depth genomics study of oil palm for selecting genes for drought tolerance.

## Figures and Tables

**Figure 1 plants-11-02317-f001:**
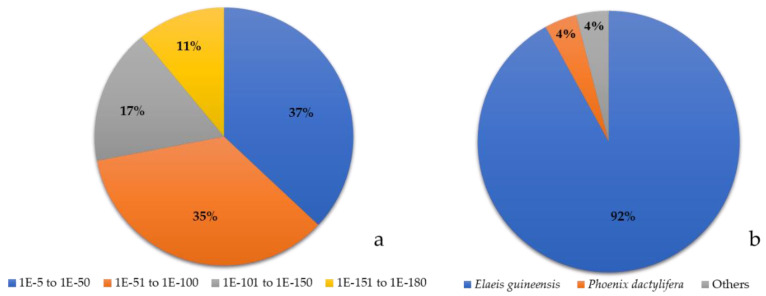
Distribution of E-value and similarly percentage of nucleotide with other plant species. (**a**). Distribution of E-value (**b**). Similarly percentage of nucleotide with other plant species.

**Figure 2 plants-11-02317-f002:**
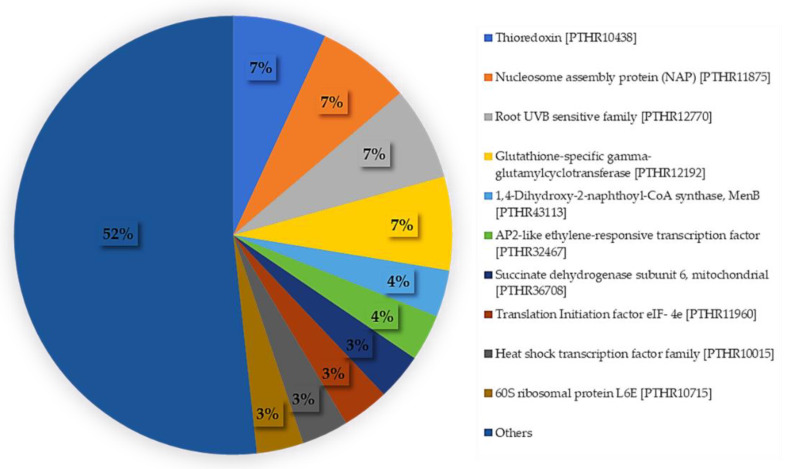
Distribution of protein function compared between the InterPro Scan and Panther databases.

**Figure 3 plants-11-02317-f003:**
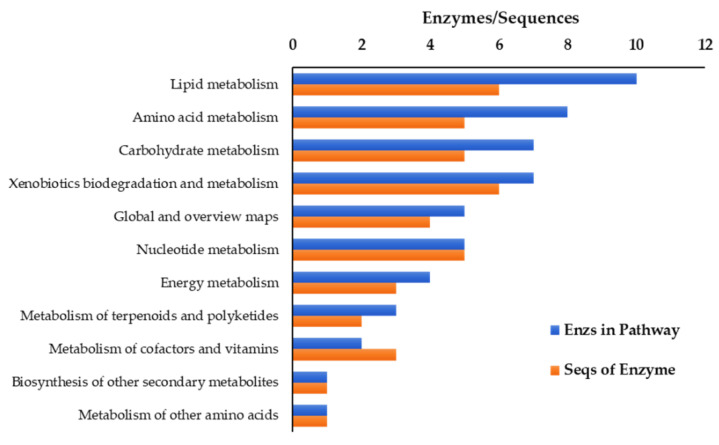
KEGG pathway analysis of drought tolerance.

**Figure 4 plants-11-02317-f004:**
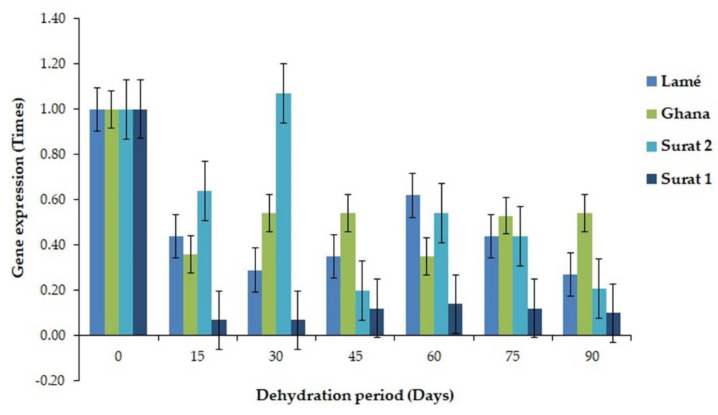
Gene expression value of PIF1 helicase, putative gene in the leaf of different oil palm varieties at varying periods of water deficit.

**Figure 5 plants-11-02317-f005:**
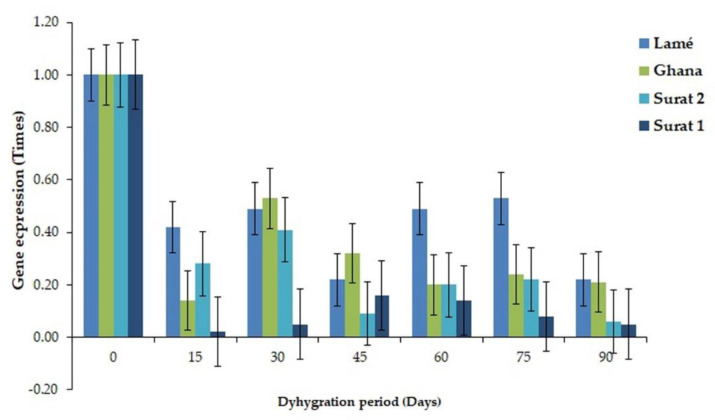
Gene expression value of transcription factor bHLH106-like gene in a leaf of different oil palm varieties at different periods of water deficit.

**Figure 6 plants-11-02317-f006:**
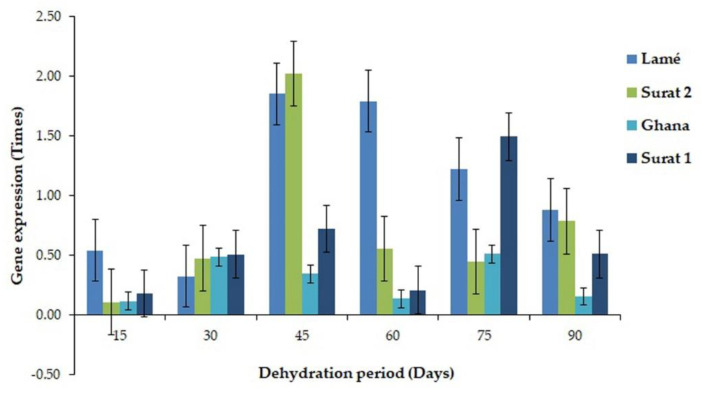
Gene expression value of *H2A* in a leaf of different oil palm varieties at different periods of water deficit.

**Figure 7 plants-11-02317-f007:**
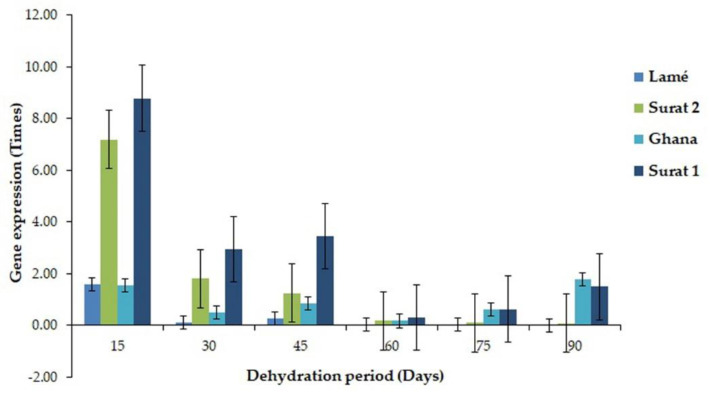
Gene expression value of *Cys* gene (cysteine proteases, *Cys*) in different oil palm root varieties at different water deficit periods.

**Figure 8 plants-11-02317-f008:**
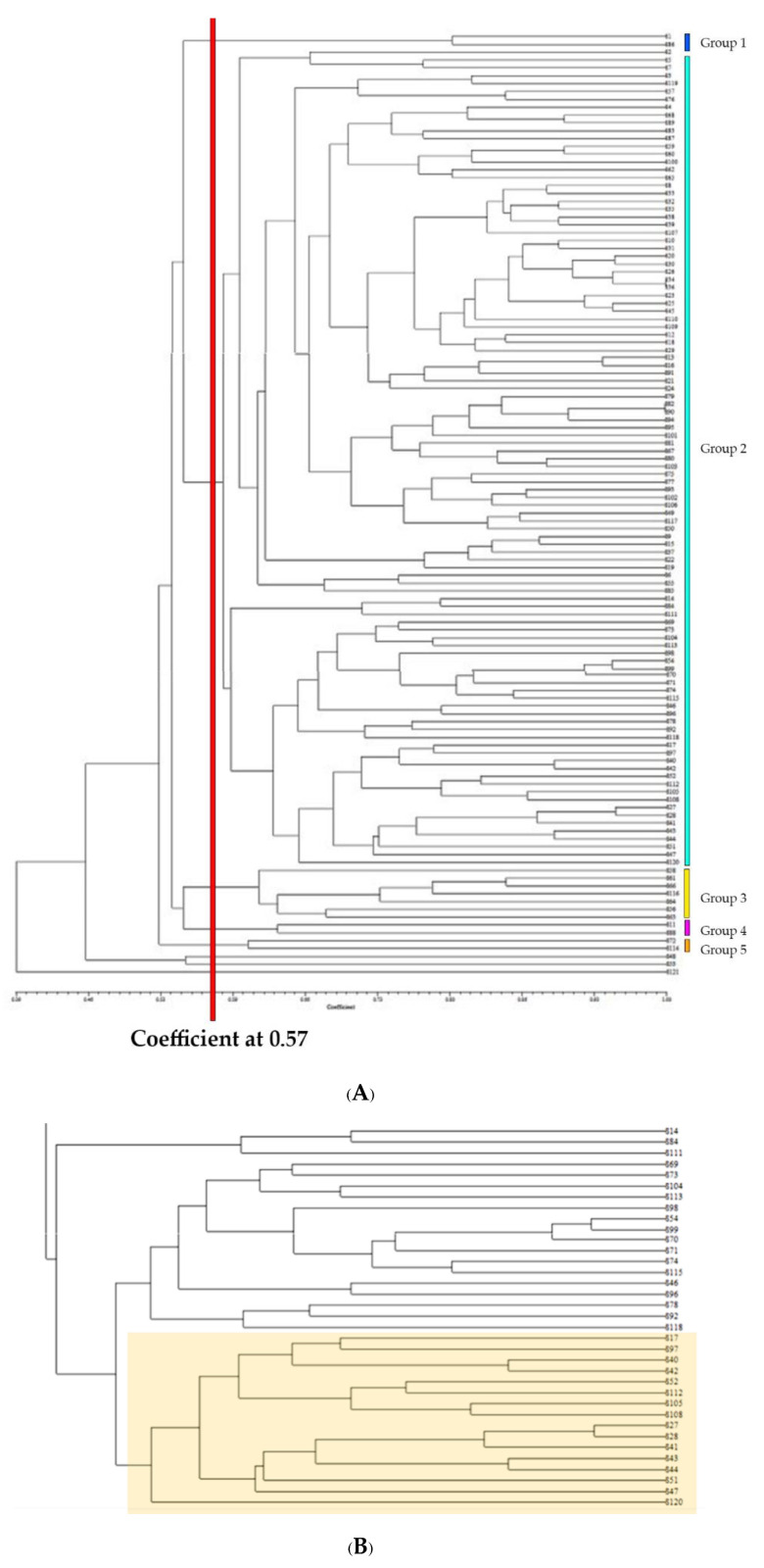
Dendrogram of genetic relationship of oil palm. (**A**). parent samples and (**B**). parent sample with a close relationship with Lamé variety (S120).

**Table 1 plants-11-02317-t001:** DDRT-PCR amplified results with primer combinations of cDNA from leaf and root of four oil palm varieties at different water deficits.

Primer	Leaf Sample			Root Sample		
	Total Bands	Polymorphism (%)	Monophism (%)	Total Bands	Polymorphism (%)	Monophism (%)
O1A2	45	86.67	13.33	51	90.20	9.80
O1A3	47	72.34	27.66	67	95.52	4.48
O1A4	74	81.08	18.92	107	97.21	2.89
O1A5	82	87.80	12.20	90	96.67	3.33
O2A2	90	80.00	20.00	95	93.68	6.32
O2A3	82	75.61	24.39	106	97.20	1.80
O2A5	135	96.30	3.70	82	98.78	1.22
O3A3	109	83.47	16.53	111	99.10	0.90
O3A4	116	79.31	20.69	105	98.10	1.90
O3A5	111	79.28	20.72	118	97.46	2.54
O4A2	117	86.32	13.68	111	94.59	5.41
O4A3	137	88.32	11.68	129	98.45	1.55
O4A5	75	94.67	5.33	76	97.37	2.63
O4A6	106	66.04	33.96	111	90.99	9.01
Total	1326	-	-	1359	-	-
Average	94.71	77.66	17.34	97.07	96.09	3.84

## Data Availability

https://bigd.big.ac.cn/gsa/browse/CRA007757, accessed on 17 August 2022.

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
