# Peer review of "Optimized Method for the Identification of Candidate Genes and Molecular Maker Development Related to Drought Tolerance in Oil Palm (Elaeis guineensis Jacq.)"

_plants, 2022, doi:10.3390/plants11172317_

Round 1
Reviewer 1 Report
The authors use the differential display reverse transcription polymerase chain reaction (DDRT-PCR) and pyrosequencing platform to get the candidate genes for drought tolerance. Then, the authors try to develop the molecular makers to identify the drought tolerance for oil palm. The aims of this study are clear. I have some suggestions for this manuscript. First, the basic information for four anchor primers and six arbitrary primers should be included either in the manuscript or the supplementary data. In the manuscript, the readers can not get any sequences or information for these primers. Second, the major mechanism of drought tolerance in this manuscript is not clear, indeed, it is hard to figure out the mechanism or pathway. I will suggest using the terms “related” or “correlated” in this manuscript. Third, the link between the DDRT-PCR and transcriptome and molecular marker development is weak in this study. The authors figure out they use 142 genes to generate the EST-SSR. Are they candidate genes for drought tolerance? If not, why did the authors put them in their manuscript? It is hard to follow in this section. Fourth, the structure of this manuscript needs to improve based on a logical attitude. It looks like the authors try to put two different parts into one manuscript in the current form. Five the English needs polish by native English speakers.
Author Response
First, the basic information for four anchor primers and six arbitrary primers should be included either in the manuscript or the supplementary data. In the manuscript, the readers cannot get any sequences or information for these primers.
Author: Already put the anchor primers and six arbitrary primers in supplementary Table 8
Second, the major mechanism of drought tolerance in this manuscript is not clear, indeed, it is hard to figure out the mechanism or pathway. I will suggest using the terms “related” or “correlated” in this manuscript.
Author: rewrite paragraph of this comment was done in discussion in line 361-401
Third, the link between the DDRT-PCR and transcriptome and molecular marker development is weak in this study. The authors figure out they use 142 genes to generate the EST-SSR. Are they candidate genes for drought tolerance? If not, why did the authors put them in their manuscript? It is hard to follow in this section.
Author: The 142 genes that was used for generate of EST-SSR are the candidate genes for drought tolerance in this experiment.
After assembly data from pyrosequencing of the two libraries, drought tolerance and drought sensitive, the obtained sequences were BLSATX with NCBI database to selected proteins related with drought tolerance. The result found 117 proteins as report on line 105-113. Then EST-SSR was done by compared repeated sequence positions and removed of replicated gene in drought library to get 142 genes used for develop of molecular marker as explained in line 114-122.
Fourth, the structure of this manuscript needs to improve based on a logical attitude. It looks like the authors try to put two different parts into one manuscript in the current form.
Author: This article used like to compare the identification of gene via DDRT-PCR and pyrosequencing methods. Both methods reveled different information but at least some genes were matched together. (Supplementary table 5)
Five the English needs polish by native English speakers.
Author: After edited follow reviewer comments, this article was sent for English proof by Best Edit & Proof.

Reviewer 2 Report
The manuscript plants-1897770, entitled “Optimized method for the identification of candidate genes and molecular maker development related to drought tolerance in oil palm (Elaeis guineensis Jacq.)” reported and discussed the results of a laboratory experiment were the drought stress gene response on oil palm was assessed. In particular, the authors compared the response of this genes expression on four different varieties taking into consideration several different genes.
In general, the manuscript and the experimental activity carried out seem to be of good quality following a strict scientific logic and according to widely used methods which have made it possible to obtain reliable results. In my opinion, only minor changes are needed before publication, regarding some typos and image quality.
Introduction: it is of good quality and gives an appropriate description of the state of the art, as well as, present the relevance of this study.
Results: are clear written.
Materials and methods: are well detailed and do not needs revisions.
Discussion: is well articulated considering each aspect of the experimental activity.
Conclusions: summarize the principal information given from this study and gives indications for further studies.
Author Response
In my opinion, only minor changes are needed before publication, regarding some types and image quality.
Author: already check for types and image quality

Round 2
Reviewer 1 Report
Supplemental 318
The authors revised their manuscript based on reviewers' comments. The last suggestion for this manuscript is to put the supplementary Figure 3 into the manuscript as Figure 8A and change the current Figure 8 into Figure 8B. The demographic pattern has information for the breeding.
Author Response
The authors revised their manuscript based on reviewers' comments. The last suggestion for this manuscript is to put the supplementary Figure 3 into the manuscript as Figure 8A and change the current Figure 8 into Figure 8B. The demographic pattern has information for the breeding.
Author : already edited based on reviewers' comments
